Immunization with tegument nucleotidases associated with a subcurative praziquantel treatment reduces worm burden following Schistosoma mansoni challenge

Rofatto Henrique K. 1 4
Araujo-Montoya Bogar O. 1 4
Miyasato Patrícia A. 2
Levano-Garcia Julio 3
Rodriguez Dunia 4
Nakano Eliana 2
Verjovski-Almeida Sergio 3
Farias Leonardo P. 4
Leite Luciana C.C. 4 lccleite@butantan.gov.br
1 Pós-Graduação Interunidades em Biotecnologia, Instituto Butantan , São Paulo , Brazil
2 Laboratório de Parasitologia, Instituto Butantan , São Paulo , Brazil
3 Departamento de Bioquímica, Instituto de Química, Universidade de São Paulo , São Paulo , Brazil
4 Centro de Biotecnologia, Instituto Butantan , São Paulo , Brazil
Hooker Bill
Electronic publication date: 2013 Apr 2
Publication date: 2013
Volume: 1
Electronic Location ID: e58
Received 2012 Nov 15; Accepted 2013 Mar 6
Copyright: © 2013 Rofatto et al.
Copyright year: 2013
Copyright holder: Rofatto et al.
License: This is an open access article distributed under the terms of the Creative Commons Attribution License, which permits unrestricted use, distribution, and reproduction in any medium, provided the original author and source are credited.
License URL: https://creativecommons.org/licenses/by/3.0/

Keywords: Schistosoma mansoni, Vaccine, SmNTPDase Apyrase ATPDase, Alkaline phosphatase, Nucleotide pyrophosphatase/phosphodiesterase SmNPP NPP, Tegument, Nucleotidases, Praziquantel

Funding: Fundação Butantan FAPESP (Fundação de Amparo a Pesquisa do Estado de São Paulo) CNPq (Conselho Nacional de Desenvolvimento Científico e Tecnológico) Supported by grants from Fundação Butantan, FAPESP (Fundação de Amparo a Pesquisa do Estado de São Paulo) and CNPq (Conselho Nacional de Desenvolvimento Científico e Tecnológico) to LCCL and SVA, and by fellowships from FAPESP to HKF, BOAM, JL and LPF. The funders had no role in study design, data collection and analysis, decision to publish, or preparation of the manuscript.

==============================
Schistosomiasis is a debilitating disease caused by flatworm parasites of the Schistosoma genus and remains a high public health impact disease around the world, although effective treatment with Praziquantel (PZQ) has been available since the 1970s. Control of this disease would be greatly improved by the development of a vaccine, which could be combined with chemotherapy. The sequencing of the Schistosoma mansoni transcriptome and genome identified a range of potential vaccine antigens. Among these, three nucleotidases from the tegument of the parasite, presumably involved in purinergic signaling and nucleotide metabolism, were proposed as promising vaccine candidates: an alkaline phosphatase (SmAP), a phosphodiesterase (SmNPP-5) and a diphosphohydrolase (SmNTPDase). Herein, we evaluate the potential of these enzymes as vaccine antigens, with or without subcurative PZQ treatment. Immunization of mice with the recombinant proteins alone or in combination demonstrated that SmAP is the most immunogenic of the three. It induced the highest antibody levels, particularly IgG1, associated with an inflammatory cellular immune response characterized by high TNF-α and a Th17 response, with high IL-17 expression levels. Despite the specific immune response induced, immunization with the isolated or combined proteins did not reduce the worm burden of challenged mice. Nonetheless, immunization with SmAP alone or with the three proteins combined, together with subcurative PZQ chemotherapy was able to reduce the worm burden by around 40%. The immunogenicity and relative exposure of SmAP to the host immune system are discussed, as key factors involved in the apparently synergistic effect of SmAP immunization and subcurative PZQ treatment.

Introduction

Schistosomes are parasitic blood flukes that cause schistosomiasis, a tropical disease that has a major public health impact in endemic countries. It affects over 200 million individuals worldwide causing more than 200,000 deaths per year, with almost 800 million people at risk of infection (Bergquist, 2002; Engels et al., 2002; Steinmann et al., 2006). Chemotherapy with Praziquantel (PZQ) is the main control strategy used; however, mass treatment does not prevent reinfection and its cumulative effects (Wilson & Coulson, 1999). Furthermore, selection of drug resistant parasites is of concern (Fallon & Doenhoff, 1995; Ismail et al., 1999). Thus, the development of a defined vaccine against schistosomiasis that could be associated with chemotherapy would contribute to the current control strategy (Bergquist, 2002).

The Schistosoma mansoni and Schistosoma japonicum transcriptomes (Hu et al., 2003; Verjovski-Almeida et al., 2003) and genomic sequencing projects (Berriman et al., 2009; Zhou et al., 2009), together with data from proteomics studies (Curwen et al., 2004; van Balkom et al., 2005; Braschi et al., 2006; Braschi & Wilson, 2006; Castro-Borges et al., 2011a; Castro-Borges et al., 2011b) have opened new opportunities for diagnosis, drug discovery and vaccine research. In several proteomic studies, three different enzymes involved in nucleotide metabolism were identified: alkaline phosphatase (SmAP), phosphodiesterase (SmNPP-5) and diphosphohydrolase (SmNTPDase). These enzymes were determined to be components of the tegument (van Balkom et al., 2005), surface membrane-associated (Braschi et al., 2006) and surface-exposed proteins accessible to biotin labeling (Braschi & Wilson, 2006). The tegument is a thin syncytial layer that covers the whole parasite, limited by a multilaminate surface membrane complex, which constitutes the major host–parasite interface (Skelly & Wilson, 2006).

In spite of the fact that SmAP has long been used as a marker for tegument membranes (Roberts et al., 1983), characterization of Alkaline Phosphatase from S. mansoni at the molecular level was performed only recently (Araujo-Montoya et al., 2011; Bhardwaj & Skelly, 2011). Both studies concluded that the enzyme was expressed throughout the parasite life cycle and showed a widespread distribution in adult worms. One study determined the surface activity of the enzyme, which was not inhibited by antibodies (Araujo-Montoya et al., 2011). The other study knocked down its expression by RNAi, which did not alter the parasite’s morphology or behavior (Bhardwaj & Skelly, 2011).

We have performed the molecular characterization of Phosphodiesterase-5 (SmNPP-5), showing its increased expression levels in the transition to intra-host stages. The surface enzymatic activity of the protein was demonstrated in living adult worms as well as its inhibition by anti-SmNPP-5 antibodies (Rofatto et al., 2009). Furthermore it was demonstrated that parasites, whose expression of the SmNPP-5 gene was suppressed at the time of host infection, were greatly impaired in their ability to establish infection (Bhardwaj et al., 2011). Additionally, the SmNTPDase enzyme has been shown to have 2 isoforms, both in the tegument, one secreted and more abundant in the syncytium and the other detected on the tegument basal and apical membranes (DeMarco et al., 2003; Levano-Garcia et al., 2007).

Surface localization is an important characteristic for vaccine candidates, to allow interaction with the host immune system. Therefore, immunization using a combination of these 3 enzymes has been proposed as a potential vaccine strategy (Braschi et al., 2006). Furthermore, it has been proposed that these enzymes may be involved in the purinergic signaling at the endothelial tissue. Since parasite movement and oviposition may cause injury and release DAMPs (damage-associated molecular pattern molecules) such as ATP, these proteins may modulate host inflammatory response by regulating the levels of ATP and ADP. Based on this hypothesis, these molecules were proposed as promising drug targets and vaccine candidates (Bhardwaj & Skelly, 2009).

Altogether these data suggest that the three tegument nucleotidases are likely to be located on the host-parasite interface, performing overlapping functions relevant for parasite survival, so the use of a cocktail of these antigens in vaccine experiments may elicit a synergistic effect. In this study, we evaluated the potential of these enzymes involved in nucleotide metabolism as vaccine candidates with and without a subcurative PZQ treatment.

Materials and Methods

Ethics statement

Animal experiments were conducted in accordance with the Brazilian Federal Law number 11.794, which regulates the scientific use of animals. All animals were handled in strict agreement with good animal practice according to the institutions guidelines for animal husbandry and all protocols were approved by the Committee of Ethics on the Use of Animals from Instituto Butantan (CEUAIB) under license 595-09.

Parasite maintenance

The S. mansoni (BH strain) entire life cycle is maintained routinely in Biomphalaria glabrata snails and hamsters at Laboratório de Parasitologia – Instituto Butantan. Cercariae were prepared for mice infection or challenge by exposing infected snails to light for 2 h to induce shedding; their number and viability were determined using a light microscope prior to infection.

Expression and purification of recombinant tegument nucleotidases

The recombinant tegument nucleotidases have been previously characterized, expressed and purified, therefore we used the hexahistidine-tag expression vectors in which the respective cDNA sequences were directionally cloned: pAE-6His vector for SmAP and SmNPP-5 (Rofatto et al., 2009; Araujo-Montoya et al., 2011) and pET 21-b vector (Novagen) for SmNTPDase (DeMarco et al., 2003). These plasmids were used to transform E. coli BL21 Star (DE3) pLys (Invitrogen), which were grown in LB medium plus ampicillin (100 µg/mL) until reaching OD600 0.6. Isopropyl-β-D-thiogalactopyranoside (Invitrogen) was added to the culture to a final concentration of 1 mM and cells were incubated for 4 h at 37 °C. Then cells were harvested by centrifugation and resuspended in 50 ml of lysis buffer (50 mM sodium phosphate pH 8.5, 0.3 M NaCl). The cell suspension was passed three times (2000 psi each) through a French press and the crude homogenate was centrifuged at 20,000×g for 30 min. The pelleted inclusion bodies were washed twice with wash buffer (lysis buffer plus 2 M urea) and finally resuspended in solubilization buffer (lysis buffer, 5 mM β-mercaptoethanol, 20 mM imidazole, 8 M urea).

The recombinant proteins were then purified by immobilized metal affinity chromatography using the Äkta Prime system (GE Healthcare) under denaturing conditions, as previously described with slight modifications (DeMarco et al., 2003; Rofatto et al., 2009; Araujo-Montoya et al., 2011). Briefly, the samples were loaded onto a 5 mL bed volume Ni2+-NTA column (GE Healthcare) pre-equilibrated with the same buffer. The columns were washed with 10 bed volumes of the equilibration buffer and then eluted with a 20–500 mM imidazole linear gradient. The main peak was pooled and the protein purity of fractions was assessed using sodium dodecyl sulfate – polyacrylamide gel electrophoresis (SDS-PAGE). Further, the elution fraction was dialyzed against Phosphate Buffer Saline pH 7.4 (PBS) and protein concentration measured with DC Protein Assay (BioRad) prior to use of these proteins (Fig. S1).

Immunization, challenge and worm recovery

Six to eight week-old female C57BL/6 mice from the Faculdade de Medicina – USP animal facility were supplied with food and water ad libitum. The animals were divided into five groups with 10 mice each: Control, SmAP, SmNPP-5, SmNTPDase and the 3 combined proteins (3Teg-Nucl). The animals were lightly anaesthetized (45 mg/kg of Ketamine and 10 mg/kg of Xylazine) before they were injected subcutaneously in the nape of the neck with 3 doses, at 15-day intervals. The SmAP, SmNPP-5 and the SmNTPDase animals were immunized with 25 µg of the respective recombinant protein mixed with Freund’s Complete Adjuvant in the first dose (Sigma) or Freund’s Incomplete Adjuvant (Sigma) in the subsequent doses. The animals immunized with 3Teg-Nucl received 25 µg of each recombinant protein with Freund’s adjuvant, while the control group was inoculated with PBS with Freund’s adjuvant using the same immunization protocol.

Fifteen days after the last dose, mice were challenged with cercariae. The animals were anaesthetized with Ketamine (90 mg/kg) and Xylazine (10 mg/kg) and exposed percutaneously for 30 min to 100 cercariae in water on their shaven abdomens by the ring method. Forty five days after percutaneous challenge, animals were euthanized with a lethal dose of urethane solution (150 mg/mL) (Sigma). Perfusion fluid (saline solution plus 500 units/L of heparin) was pumped into the aorta artery, and perfused worms were collected from the hepatic portal vein and counted using a stereomicroscope.

The protection was calculated by comparing the number of worms recovered from each vaccinated group with the control group, using the formula: Protection level (%)=[(WRCG-WREG/WRCG)]∗100

where WRCG = worms recovered from control group and WREG = worms recovered from experimental group. To evaluate the liver-trapped egg burden, a piece of the liver from each mouse was removed, weighed, digested and homogenized for 1 h at 37 °C in 5 mL of 10% KOH. The number of eggs per gram of liver was compared to the control group (Cardoso et al., 2008). All these data were statistically compared by ANOVA followed by Dunnett’s test and a ρ ≤ 0.05 was considered significant.

Measurement of specific anti-nucleotidases antibody levels

Mice were bled from the retro orbital plexus one day before cercariae challenge (day 44) and one day before mouse perfusion (day 89). The blood was processed and the sera collected were used to perform indirect ELISA assays to confirm the levels of specific anti-nucleotidases total IgG, IgG1 and IgG2a. Maxisorp 96-well microtiter plates (Nunc) were coated with 5 µg/mL of recombinant SmAP, SmNPP-5 or SmNTPDase in carbonate-bicarbonate buffer, pH 9.6 for 18 h at 4 °C, then blocked for 1 h at 37 °C with 200 µL/well phosphate buffer saline, pH 7.2 with 0.05% Tween-20 plus 10% fetal bovine sera (PBS-T). The serum of each animal was serially diluted starting at 1:50, and incubated for 1 h at 37 °C. Plate-bound antibodies were detected by goat anti-mouse IgG, IgG1 or IgG2a (Southern Biotech) diluted in PBS-T 1:10,000, 1:1,000 or 1:1,000, respectively. Finally the plates were incubated with peroxidase-conjugated rabbit anti-goat IgG diluted 1:20,000 for 1 h at 37 °C and color reaction was developed by incubation of o-Phenylenediamine dihydrochloride (Sigma) in citrate buffer, pH 5.0 plus 0.04% H2O2 for 15 min and stopped with 4 M sulfuric acid. The plates were read at 492 nm in an ELISA plate reader (Labsystems). As background of ELISA assays presented an OD492 around 0.05, we selected the dilutions closer to OD492 0.1 to correlate with standard curves, which were generated using mouse IgG, IgG1, and IgG2a (Southern Biotech). Statistical comparisons were performed with ANOVA followed by a Tukey’s pairwise comparison. A ρ value ≤0.05 was considered statistically significant.

Cytokine analysis

Cytokine experiments were performed using splenocyte cultures from individual mice (4 animals for each group) immunized with isolated or combined tegument nucleotidases as described above. Splenocytes were isolated from macerated spleens 15 days after the third immunization, and washed twice with sterile PBS. After washing, the cells were adjusted to 1 × 106 cells in 1 ml of RPMI 1640 medium (Invitrogen) supplemented with 10% fetal bovine sera, 100 U/mL of penicillin G sodium, 100 µg/mL of streptomycin sulfate, 250 ng/mL of amphotericin B and polymyxin B (30 µg/ml). Splenocytes were restimulated in culture with each recombinant nucleotidase (5 µg/mL) or concanavalin A (ConA) (5 µg/ml) for 48 h in an incubator at 37 °C with 5% CO2. Culture supernatants were collected for IFN-γ and IL-4 analysis using an ELISA kit according to the manufacturer’s directions (Peprotech).

RNA from splenocytes was extracted and purified using Trizol (Invitrogen) and cDNA was synthesized using the Superscript III Reverse Transcriptase kit (Invitrogen). RT-PCR of the sample was performed in a thermal cycler under the manufacturer instructions using random hexamers. The following genes were analyzed using Taqman Gene Expression Assays for gene quantification: β-actin (4352933E) as a housekeeping gene, MyD88 (Mm00440338_m1), NF-κB1 (Mm00476361_m1), NF-κB2 (Mm00479807_m1), IL-4 (Mm00445259_m1), IL-5 (Mm00439645_m1), IL-10 (Mm00439616_m1), IL-12p40 (Mm01288992_m1), IL-13 (Mm00434206_g1), IL-17 (Mm00439619_m1), IFN-γ (Mm00801778_m1), TNF-α (Mm00443258_m1) and TGF-β (Mm00441724_m1). The level of β-actin was used to normalize the amounts of assayable RNA in each sample and quantitation of relative differences in expression were finally calculated using the comparative ΔΔCt method (Livak & Schmittgen, 2001), using the expression levels of PBS-immunized animals as baseline controls. Statistical comparisons were performed with ANOVA followed by a Tukey’s pairwise comparison. A ρ value ≤0.05 was considered statistically significant.

Analysis of worm morphology after subcurative Praziquantel chemotherapy

Six to eight week-old female C57BL/6 mice were divided into control and subcurative PZQ treated groups containing 4 animals each and mice were infected with cercariae as described above. The subcurative PZQ treated animals received two doses of 150 mg/kg of PZQ by gavage, 35 and 37 days after infection respectively; saline solution was administered for control animals (adapted from Doenhoff, Modha & Lambertucci, 1988). Two hours after the second PZQ dose, animals were perfused to recover parasites as described above. Recovered worms were fixed in Formalin–Acetic acid–Ethanol solution and analyzed with a laser confocal microscope (LSM 510 META, Zeiss) at 488 nm wavelength of excitation and 500–550 nm wavelength of emission. The tubercle numbers from control and treated groups were compared with Student’s T-test (based on Moraes et al., 2011).

Results and discussion

Humoral immune response induced by immunization with tegument nucleotidases

In order to investigate the immunogenicity of the tegument nucleotidases, groups of C57BL/6 mice were immunized with either the isolated proteins or with a combination of the 3 tegument nucleotidases (3Teg-Nucl), in a 3-dose schedule with 2 weeks intervals, including the respective controls. The animals were challenged 2 weeks after the last dose and perfused after 45 days. Sera were collected before challenge (day 44) and before perfusion (day 89) (Fig. 1A). Total IgG, IgG1 and IgG2a antibody levels against SmAP, SmNPP-5 and SmNTPDase were determined. Anti-SmAP total IgG levels were higher than anti-SmNPP-5 and anti-SmNTPDase levels before and after cercaria challenge, irrespective of whether they were administered isolated or combined (Figs. 1B and 1C). No difference was observed in Total IgG levels of the groups before and after challenge (Figs. 1B and 1C). All groups differ from the respective controls, which presented negligible values (ρ < 0.001; data not shown). This data may indicate that these nucleotidases may not be accessible to the host immune system to boost the antibody levels.

Figure 1 Standard immunization schedule and total IgG levels induced by immunization with tegument nucleotidases.

(A) Immunization, bleeding, challenge and mouse perfusion schedule. (B) Specific total IgG induced by immunization with isolated nucleotidases before and after challenge. (C) Specific total IgG induced by immunization with combined nucleotidases (3Teg-Nucl) before and after challenge. The bars are Mean ± SEM; ∗ = ρ ≤ 0.05.

Immunization of mice with SmAP induced the highest IgG1/IgG2a ratio, indicating a Th2 predominance, which decreased by half following challenge. Anti-SmNPP-5 IgG1/IgG2a ratio also decreased following challenge, but was higher than SmNTPDase before challenge. On the other hand, SmNTPDase ratio was the lowest and was not altered by challenge (Figs. 2A–2C). When the proteins were combined, the IgG1/IgG2a ratio of most groups seemed higher and not to be altered by challenge (Figs. S2A–S2C).

Figure 2 IgG1 and IgG2a levels induced by immunization with isolated tegument nucleotidases.

(A) Specific IgG1 and IgG2a levels induced by immunization with SmAP before and after challenge. (B) Specific IgG1 and IgG2a levels induced by immunization with SmNPP-5 before and after challenge. (C) Specific IgG1 and IgG2a levels induced by immunization with SmNTPDse before and after challenge. The bars are Mean ± SEM. The numbers over the bars are the IgG1/IgG2a ratios.

Cellular immune response induced by immunization with tegument nucleotidases

The cellular immune response induced by the immunization with the tegument nucleotidases was evaluated in groups of animals immunized with either the isolated proteins or with a combination of the 3 tegument nucleotidases (3Teg-Nucl) in a 3-dose schedule with 2-week intervals. After 2 weeks, the splenocytes were collected and the cellular immune response was evaluated in the same cells by ELISA and real-time RT-PCR after in vitro stimulation with each protein. The ELISA results revealed that the levels of the basic Th1 and Th2 cytokines (IFN-γ and IL-4) induced by the tegument nucleotidases, alone or combined, were not detectable, in spite of a strong positive response with ConA (results not shown).

Analysis of the real-time RT-PCR results showed that when the animals were immunized with the isolated proteins, SmAP induced extremely high expression levels of the inflammatory cytokine, TNF-α, and together with SmNPP-5 showed higher expression of the Th17 cytokine, IL-17. However, SmNPP-5 was also characterized by expression of the regulatory cytokine, TGF-β, (Fig. 3). SmNTPDase showed induction of relatively lower levels of cellular immune responses. When the proteins were combined, similar results were obtained, with SmAP inducing high levels of TNF-α, associated with increased levels of IL-17. SmNPP-5 in combination with the other nucleotidases also produced increased levels of IL-17, however, also counter-balanced by significant levels of TGF-β (Fig. S3). Interestingly, no differences in expression of characteristic Th2 cytokines, such as IL-4, IL-5 and IL-13 were observed, either induced by the immunization with isolated or combined proteins, although the antibody isotype profiles, which showed a predominance of IgG1, were consistent with a Th2 response.

Figure 3 Cellular immune response induced by immunization with isolated tegument nucleotidases and in vitro reestimulation with each protein evaluated by qPCR.

The bars are Mean ± SEM; ∗ = ρ ≤ 0.05.

Evaluation of worm burden and liver-trapped eggs induced by standard immunization protocol with tegument nucleotidases

The protective potential of the tegument nucleotidases was evaluated by immunization of the animals with either the isolated proteins or with a combination of the 3 proteins (3Teg-Nucl). Mice were challenged 2 weeks after the last dose and after 45 days the worms were recovered from the mesenteric vein of immunized and control animals. A piece of liver was also collected to evaluate liver-trapped egg counts. It can be observed that immunization with either the isolated or combined proteins did not reduce the worm burden of immunized mice nor did it reduce the amount of liver-trapped eggs (Figs. 4A and 4B). This is one representative of two independent experiments.

Figure 4 Evaluation of tegument nucleotidases as vaccine candidates by standard immunization protocol.

(A) Worm burden dispersion of mice immunized with tegument nucleotidaes and challenged with live cercariae; the lines represent the Means. (B) Liver-Trapped Eggs from mice immunized with tegument nucleotidases, the bars are the Mean ± SEM.

Humoral immune response induced by immunization with tegument nucleotidases associated with Praziquantel subcurative treatment

Since it has been shown that treatment of S. mansoni-infected mice with subcurative doses of PZQ would induce alterations in the parasites’ tegument (Liang et al., 2002), we initially evaluated the integrity of the parasites surfaces by autofluorescence and their survival following the subcurative doses of PZQ. Under the conditions used in our experiments, the parasites recovered from mice after PZQ treatment visually inspected by confocal microscopy did not show significant morphological alterations, although their tegumental tubercles were slightly less defined, but not completely destructured. The parasites’ survival was not impaired by the PZQ treatment either (Fig. S4). Therefore, we evaluated the combined effect of immunization with tegument nucleotidases associated with subcurative doses of PZQ (Fig. 5A). As in the previous experiment, SmAP and SmNTPDase showed comparable levels of Total IgG, higher than SmNPP-5, either as isolated proteins or administered as a combination, 3Teg-Nucl (Figs. 5B and 5C). It is interesting to note that the total anti-SmAP IgG before challenge was 6,600 µg/mL in the first experiment, which was reduced to 4,350 µg/mL before perfusion (a reduction of 34%, although not statistically significant, Fig. 1B). When the proteins were combined the reduction was even larger (44%, although still not significant, Fig. 1C). However, when the challenge was performed in the presence of PZQ, either with the isolated proteins or combined, the level of anti-SmAP was comparable (Figs. 5B and 5C). This data may indicate that the protein could be more accessible to the immune system after PZQ treatment, as previously reported (Fallon et al., 1994).

Figure 5 Schedule for immunization with tegument nucleotidases associated with praziquantel subcurative treatment and induction of total IgG.

(A) Immunization, bleeding, challenge, subcurative praziquantel treatment and perfusion schedule. (B) Specific total IgG induced by immunization with isolated nucleotidase before and after challenge and praziquantel therapy. (C) Specific total IgG induced by immunization with combined nucleotidases (3Teg-Nucl) before and after challenge and praziquantel therapy. The bars are Mean ± SEM; ∗ = ρ ≤ 0.05.

The sera of mice immunized with tegument nucleotidases and treated with subcurative doses of PZQ were also evaluated as to the IgG1/IgG2a ratio, confirming the Th2 predominance of the immune response before challenge and administration of the PZQ doses at day 44, as previously determined (Figs. 6A–6C). Following the challenge and PZQ treatments, the immune response for most of the groups displayed a reduction in the IgG1/IgG2a ratio, indicating a shift towards Th1 characteristics (Figs. 6A–6C, and Fig. S5). This is probably due to the effect of the drug on the parasites, which altered its interaction with the host immune system. In both experiments with and without PZQ treatment, it is clear that before challenge, SmAP and SmNTPDase were more immunogenic than SmNPP-5, inducing higher levels of antibodies, mostly IgG1, and after challenge, there is a general decrease in the levels of IgG1 and IgG2a against all antigens.

Figure 6 IgG1 and IgG2a levels induced by immunization with isolated tegument nucleotidases associated with praziquantel subcurative treatment.

(A) Specific IgG1 and IgG2a levels induced by immunization with SmAP before challenge and after praziquantel treatment. (B) Specific IgG1 and IgG2a levels induced by immunization with SmNPP-5 after praziquantel treatment and before challenge. (C) IgG1 and IgG2a levels induced by immunization with SmNTPDse before challenge and after praziquantel treatment. The bars are Mean ± SEM. The numbers over the bars are the IgG1/IgG2a ratios.

Reduction in worm burden and liver-trapped eggs induced by immunization with tegument nucleotidases associated with Praziquantel subcurative treatment

It has been demonstrated that chemotherapy with PZQ interacts synergistically with host immune responses for parasite elimination (Brindley & Sher, 1987; Fallon & Doenhoff, 1995). Therefore we combined immunization and challenge protocols with a subcurative treatment, expecting that chemotherapy would enhance the efficacy of the immunizations. Groups of animals immunized with tegument nucleotidases were challenged with live cercaria and received two subcurative doses of PZQ after 35 and 37 days. Adult worms were recovered from the mesenteric veins of perfused mice, 45 days after challenge (Fig. 5A). In the first experiment, immunization with the combined proteins, 3Teg-Nucl, followed by PZQ treatment, induced a 41% reduction in worm burden, as compared to the Control also treated with PZQ (Fig. 7A). Furthermore, this protocol also induced a 54% reduction in egg deposition (Fig. 7C).

Figure 7 Evaluation of the protective potential of tegument nucleotidases associated with praziquantel subcurative treatment.

(A) and (B) Worm burden dispersion and percentage of protection from mice immunized with tegument nucleotidases and treated with subcurative chemotherapy; the lines represent the Means. (C) and (D) Liver-Trapped Eggs reduction induced by immunization with tegument nucleotidases associated with praziquantel subcurative treatment; the bars are the Mean ± SEM. ∗ = ρ ≤ 0.05.

In order to determine the contribution of each nucleotidase to the observed protection, we performed a second challenge experiment, also including additional groups of animals immunized with the isolated proteins. In this experiment, it was possible to confirm the protective effect of 3Teg-Nucl, which displayed a worm burden reduction of 46%, and to establish that the main protein contributing to worm burden reduction was SmAP, which conferred a reduction of 41% (Fig. 7B). The liver-trapped egg count also indicated its contribution, although statistical significance was not reached (Fig. 7D).

SmAP is clearly more immunogenic than the other tegument nucleotidases, as suggested by the higher antibody and cytokine expression levels induced, which could be one of the factors leading to its higher protective potential. The high IgG1/IgG2a ratio induced by immunization with SmAP indicates the induction of a Th2-predominant immune response. However, the cytokine profile, with low expression of Th2 cytokines, and higher expression of pro-inflammatory cytokines, suggests a more mixed response. The results indicate that the inflammatory cytokine, TNF-α, associated with increased levels of IL-17, could also be involved in the protective mechanism. Despite the evidence that induction of a Th1 immune response would be more effective in preventing schistosomiasis in mice (Hoffmann et al., 1999; Cardoso et al., 2008; Teixeira de Melo et al., 2010), it is well documented that a Th2 or a mixed immune response can also reduce parasite worm burden following infection, if it is strong enough (Hoffmann et al., 1999; El Ridi & Tallima, in press; Martins et al., 2012).

Although we have previously shown that anti-SmNPP-5 antibodies induced by this recombinant protein inhibit the activity of SmNPP-5 in live parasites (Rofatto et al., 2009), the recombinant protein was not in the native conformation, i.e., not properly folded. Furthermore, our results showed that immunization of mice with SmNPP-5 induced lower humoral immune responses. On the other hand, the T-cell immune response that was induced was counter-balanced by a strong regulatory response, characterized by TGF-β. Moreover, it has been demonstrated that immunization with NPP-5 from S. japonicum, resulted in a partial worm burden reduction following a subsequent challenge (Zhang et al., 2012a). Therefore it is possible that a properly folded protein would show a different result. SmNTPDase was more immunogenic than SmNPP-5, displaying higher levels of antibodies, but showed lower induction of cellular immune responses as determined by the expression levels of the evaluated cytokines. Hence, the immune responses induced by the different nucleotidases may justify the different protective efficacies against challenge.

Although Bhardwaj & Skelly (2009) hypothesized that the schistosoma tegument nucleotidases function in the human bloodstream, their real physiological functions and biological substrates remain uncertain. As we could not verify any synergistic effect following immunization with the combined tegument nucleotidases, it is possible that these proteins may not share functions/substrates nor be physiologically correlated. Furthermore, suppression of SmNPP-5 expression impairs parasite infection (Bhardwaj et al., 2011), while suppression of SmAP expression has no effect on cultured parasites (Bhardwaj & Skelly, 2011); these data suggest a more vital function for SmNPP-5 than for SmAP on parasite survival. However, our results showed that only immunization with SmAP associated with PZQ subcurative treatment leads to worm burden reduction. Based on these data it seems that the efficacy of SmAP immunization may not be related to a more relevant function of this protein.

Even though more immunogenic, immunization with SmAP was only effective in reducing worm burden when associated with PZQ treatment; this may be due to a non-specific parasite injury, which involves some kind of alteration of the parasite’s immune evasion mechanisms (Brindley & Sher, 1987). Alternatively, this effect could be specific for SmAP in a synergistic way, since it has been demonstrated that PZQ treatment promotes exposure of a few tegument antigens, among them SmAP (Brindley et al., 1989; Fallon et al., 1994). Furthermore, of the three proteins, it is probably the most exposed on the surface of the parasite, since it is the only protein identified by proteomics in the fraction of proteins released following treatment of live parasites with phosphatidylinositol-phospholipase C (Castro-Borges et al., 2011a; Castro-Borges et al., 2011b). Therefore, the level of exposure of the protein to the immune system would be another explanation for the synergistically protective efficacy of SmAP. In this case, the use of subcurative PZQ treatment was a valuable tool to make this important characteristic more evident.

These results reinforce the perspective of combining immunization strategies with chemotherapeutic treatments. Despite the fact that good vaccine candidates should reduce worm burden alone, our experiments were designed to allow visualization of the synergistic effect of immunization and chemotherapy on the parasites. Similar approaches have been tested in the past, using parasite extract as antigens (Fallon & Doenhoff, 1995) and have recently been proposed again for other recombinant tegument proteins for S. japonicum (Zhang et al., 2012b). Although these conditions will not be reproduced in human treatment, it is probable that vaccine administration will occur concurrent to chemotherapy treatment. Therefore, it is interesting to envisage and evaluate different schedules of immunizations and chemotherapeutic treatments, aiming to select the more appropriate to be employed in human trials.

Conclusion

We evaluated three recombinant tegument nucleotidases from S. mansoni that have been extensively proposed as vaccine antigens in the literature. SmAP was the most immunogenic of them, inducing a mixed Th2 humoral immune response with an inflammatory cellular immune response, characterized by high TNF-α and IL-17 production. Although it was not protective alone in the formulation used, it effectively reduced worm burden, when associated with a subcurative PZQ treatment. Both SmNPP-5 and SmNTDPase showed lower immunogenicity in the presentation forms used; SmNPP-5 displayed a regulatory cellular immune response inducing low levels of antibodies, while SmNTDPase induced a Th2-predominant humoral immune response. Our data suggested that SmAP is the more promising tegument nucleotidase as vaccine candidate and advocates for additional investigations, from improving its expression and folding, through to its evaluation with different adjuvants and immunization protocols. We also consider that SmAP may be useful in a combination with other protective antigens, particularly if the other proteins induce an immune response that targets the forming tegument after infection. Finally, our studies highlight the importance of investigating vaccine candidates in different immunization schedules combined with chemotherapeutic agents.

Supplemental Information

Fig. S1 SDS-PAGE of recombinant Schistosoma mansoni nucleotidases.

MW – Protein molecular weight marker: 120, 85, 50, 35 and 20 kDa. 01 – SmAP; 02 – SmNPP-5; 03 – SmNTPDase.

Click here for additional data file.

Fig. S2 IgG1 and IgG2a levels induced by immunization with combined (3Teg-Nucl) tegument nucleotidases.

(A) Specific IgG1 and IgG2a levels against SmAP before and after challenge. (B) Specific IgG1 and IgG2a levels against SmNPP-5 before and after challenge. (C) Specific IgG1 and IgG2a levels against SmNTPDse before and after challenge. The bars are Mean ± SEM. The numbers over the bars are the IgG1/IgG2a ratios.

Click here for additional data file.

Fig. S3 Cellular immune response induced by immunization with combined tegument nucleotidases (3Teg-Nucl) and in vitro reestimulation with each protein evaluated by qPCR.

The bars are Mean ± SEM; ∗ = ρ ≤ 0.05.

Click here for additional data file.

Fig. S4 Evaluation of Praziquantel subcurative chemotherapy in vivo.

(A) and (B) Worms morphology analysis by confocal microscopy of worms perfused 2 h after second dose of Praziquantel from control and treated mice, respectively. (C) and (D) Tegumental tubercles analysis by confocal microscopy of worms perfused 2 h after second dose of Praziquantel from control and treated mice, respectively. (E) Tegumental tubercles number from worms perfused 2 h after second dose of Praziquantel treatment; the bars are the Mean ± SD. (E) Worm burden dispersion of mice perfused 45 days after infection treated or not with subcurative doses of Praziquantel; the lines represent the Means.

Click here for additional data file.

Fig. S5 IgG1 and IgG2a levels induced by immunization with combined tegument nucleotidases (3Teg-Nucl) associated with Praziquantel subcurative treatment.

(A) Specific IgG1 and IgG2a levels against SmAP before challenge and after Praziquantel treatment. (B) Specific IgG1 and IgG2a levels against SmNPP-5 before challenge and after Praziquantel treatment. (C) IgG1 and IgG2a levels against SmNTPDse before challenge and after Praziquantel treatment. The bars are Mean ± SEM. The numbers over the bars are the IgG1/IgG2a ratios.

Click here for additional data file.

We thank Alexsander Seixas de Souza from Instituto Butantan for confocal microscopy (FAPESP 00/11624-5) imaging support.

Additional Information and Declarations

Competing Interests

Author Contributions

Animal Ethics

LCCL is an Academic Editor for PeerJ. The other authors state that there are no competing interests.

Henrique K. Rofatto conceived and designed the experiments, performed the experiments, analyzed the data, wrote the paper.

Bogar O. Araujo-Montoya performed the experiments, contributed reagents/materials/analysis tools.

Patrícia A. Miyasato and Dunia Rodriguez performed the experiments.

Julio Levano-Garcia, Eliana Nakano and Sergio Verjovski-Almeida contributed reagents/materials/analysis tools.

Leonardo P. Farias conceived and designed the experiments, performed the experiments.

Luciana C.C. Leite conceived and designed the experiments, analyzed the data, wrote the paper.

The following information was supplied relating to ethical approvals (i.e. approving body and any reference numbers):

Committee of Ethics on the Use of Animals from Instituto Butantan (CEUAIB) - license number 595-09.

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
