# Peer review of "Immunization with tegument nucleotidases associated with a subcurative praziquantel treatment reduces worm burden following Schistosoma mansoni challenge"

_PeerJ, doi:10.7717/peerj.58_

## Round 0.1 · original submission · Major Revisions

The authors should respond to each of the points raised by Reviewer 1 and myself (below).

Suggested grammatical corrections

Abstract
Schistosomiasis is a debilitating disease caused by flatworm parasites of the Schistosoma genus and remains a high public health impact disease around the world, although effective treatment with Praziquantel (PZQ) has been available since the 1970s. Control of this disease would be greatly improved by the development of a vaccine, which could be combined with chemotherapy. The sequencing of the Schistosoma mansoni transcriptome and genome identified a range of potential vaccine antigens. Among these, three nucleotidases from the tegument of the parasite, presumably involved in purinergic signaling and nucleotide metabolism, were proposed as promising vaccine candidates: an alkaline phosphatase (SmAP), a phosphodiesterase (SmNPP-5) and a diphosphohydrolase (SmNTPDase). Herein we evaluate the potential of these enzymes as vaccine antigens, with or without subcurative PZQ treatment. Immunization of mice with the recombinant proteins alone or in combination demonstrated that SmAP is the most immunogenic one of the three. It induced the highest antibody levels, particularly IgG1, associated with an inflammatory cellular immune response characterized by high TNF-α and a Th17 response, with high IL-17 expression levels. Despite the specific immune response induced, immunization with the isolated or combined proteins did not reduce the worm burden of challenged mice. Nonetheless, immunization with the three proteins combined or with SmAP alone together with subcurative PZQ chemotherapy was able to reduce the worm burden by around 40%. The immunogenicity and relative exposure of SmAP to the host immune system are discussed, as are key factors involved in the apparently synergistic effect of SmAP immunization and subcurative PZQ treatment.

line 77 “…expressed throughout the parasite life cycle and showed widespread distribution in adult…”

Questions regarding data and analysis

1. The authors should provide more detail on the purity of the recombinant antigens used as immunogens. At a minimum, the SDS-PAGE gels used to assess purity (line 149) should be shown (as a supplementary figure if preferred) and the method used to measure protein concentration added to the Methods section.

2. Recombinant proteins were plated on microtiter plates in carbonate buffer, but they were eluted using an imidazole gradient (in what buffer – “equilibration buffer”? Is that the same as the solubilization buffer?). Was buffer exchange performed or were the proteins simply diluted into the plating buffer?

3. Why was retro-orbital bleeding chosen over (e.g.) tail bleeding? Why was urethane injection chosen over (e.g.) sodium pentobarbital or cervical dislocation?

4. line 264: “data not shown” – please show this data, in a supplementary figure if preferred.

5. line 307: “one representative of two independent experiments” – it looks as though both experiments are shown in Fig 4, since there are 9 or 10 datapoints for each worm burden result and an error bar is shown for the egg counts. Is this the case? If not, please show both experiments.

6. line 325: Neither a 34% nor a 44% reduction in total Anti-SmAP IgG reached statistical significance; what level of reduction would be required to reach significance, and was this addressed in the experimental design (i.e. choice of animal numbers)?

7. The authors should discuss why, following cercarial challenge, IgG levels generally fall in the immunized animals – would a challenge with an immunogen against which the immune system was primed not usually be expected to increase the titer?

8. Section 3.5: references to Figs 7A, 7B, 7C and 7D appear to be mixed up, e.g. line 358 Fig 7B shows worm burden not egg numbers.

9. It’s not clear from the text whether the Control shown in Fig 7 is PZQ treated or not. Please clarify and explain why you did not use two Control groups, with and without PZQ treatment (both unimmunized).

10. When total IgG levels and the combined levels of IgG1 and IgG2A are compared, the numbers do not add up cleanly. I understand that different detection antibodies were used for all three EIAs but if the results are expressed as µg Ig per ml one would expect ballpark agreement. For instance:

Fig 1A, day 44 anti-SmAP total IgG = ~6500 µg/ml
Fig 2A, day 44 anti-SmAP IgG1 = ~4500 and IgG2a = ~200 µg/ml
--> what explains the ~2000 µg/ml difference?

Fig 1C, day 44 total anti-SMNTPDase IgG = ~3500 µg/ml
Fig 2C, day 44 anti-SMNTPDase IgG1 = ~250 and IgG2a = ~100 µg/ml
--> what explains the >3000 µg/ml difference?

Fig 5B, day 44 total anti-SMNTPDase IgG = ~5000 µg/ml
Fig 6C, day 44 anti-SMNTPDase IgG1 = ~100 and IgG2a = ~40 µg/ml
--> what explains the >4800 µg/ml difference?

What are normal mouse serum levels of IgG (total and subtypes), and what level of variation is normal? It does not seem that subtypes other than 1 and 2a could account for the discrepancy – or if they do, perhaps that is significant?

The authors should address this issue because if it is not possible to measure the IgG levels accurately and absolutely (in µg/ml) then the results should be expressed in arbitrary units and comparisons can only be made between samples tested with the same detection antibody.

11. Figure 3: was a baseline established for expression of each cytokine in unstimulated splenocytes? Is there a control that could be used to show strong upregulation, analagous to the use of bacterial LPS in T cell proliferation assays?

Reviewer 1 ·

Basic reporting

There are a few errors of English e.g. in the Abstract the word "the" should appear before "schistosoma" and before "1970s". Later, "immunization" is more correct than "immunizations" in the abstract. On line 122: "in" not "on".

Experimental design

Generally fine. However, the authors must show us the quality of the immunogens. These are key to the entire work. Gels displaying the purity of the recombinants should be presented in the text (or as supplementary data).

Validity of the findings

Generally good. However, more should be stated about the control groups. What was the background like, throughout? Were control values equivalent for all 3 immunogens? In the data shown, are the control values subtracted?

Line 196: many dilutions were tested but results derived using only one are shown. How was this one chosen? Comment or show the rest?

Line 200: define “OPD”

Was the experiment whose results are shown in fig 2 carried out only once/using 1 pooled sample? If yes, how representative are the data? If not, show the variation detected. Put another way, how can we tell if the changes in the ratios reported at day 44 v day 89, (e.g. on line 266) are really meaningful?

Why was day 15 after the last immunization chosen as the time point to measure cytokine mRNA levels? Likewise why a 48 h (versus 24 or 72 h?) time point after stimulation?

Line 317: What exactly does it mean to say that parasites' teguments were not “significantly” altered. What alterations were detected?

Additional comments

Explain the observation that the isotype data suggest a Th2-biased response but the cytokine data are not consistent with this? Discuss.

Discuss the significance, if any, of the the changes in the ratios reported at day 44 v day 89. What drives these changes?

Line 358: Fig 7B should be 7C (or change the figures). (Same issue on line 364.)

Reviewer 2 ·

Basic reporting

No

Experimental design

The study design is well structured.

Validity of the findings

The data are robust, statistically sound, and all speculations were welcomed.

Additional comments

This study is certainly of interest to investigate the protective effect of the vaccine antigens, in mouse models, against Schistosoma mansoni infections. The findings are important to those with closely related research interests. Before publication, the article needs some language corrections, notably the format of the references should be revised to meet the requirement of the journal following the guidelines.

---

## Round 0.2 · Minor Revisions

Minor details only:

1, typographical errors:
abstract:
"Herein we evaluate" (not evaluateD)
"...most immunogenic of the three" (not "ONE of the three")
"Nonetheless, immunization..." (not immunizationS)
line 206, "...presented an OD492" (not anD)

2. OPTIONAL -- line 309, suggest expanding the sentence to explain a bit more thoroughly, as per response to Rev #1 -- e.g. "...although the antibody isotype profiles, which showed a predominance of IgG1, were consistent with a Th2 response."

3. OPTIONAL -- regarding "data not shown" -- there is no space limitation in an electronic journal! I leave the choice to the authors, but strongly encourage them to include ALL data mentioned in the text, as supplementary figures or tables where appropriate. There's no such thing as too much data -- those readers who don't need/want it can skip it.

---

## Round 0.3 · accepted · Accept

It's been a pleasure; my apologies for the delays, which I assure you were my fault and nothing to do with PeerJ!